# C-EVAL: A Multi-Level Multi-Discipline Chinese Evaluation Suite for Foundation Models

**Yuzhen Huang**[*1]   **Yuzhuo Bai**[*2]   **Zhihao Zhu**[1]   **Junlei Zhang**[1]   **Jinghan Zhang**[1]
**Tangjun Su**[1]   **Junteng Liu**[1]   **Chuancheng Lv**[2]   **Yikai Zhang**[1]   **Jiayi Lei**[1]
**Yao Fu**[3]   **Maosong Sun**[2]   **Junxian He**[†4]

[1]Shanghai Jiao Tong University   [2]Tsinghua University   [3]University of Edinburgh
[4]The Hong Kong University of Science and Technology
ceval.benchmark@gmail.com
https://cevalbenchmark.com

## Abstract

New NLP benchmarks are urgently needed to align with the rapid development of large language models (LLMs). We present C-EVAL , the first comprehensive Chinese evaluation suite designed to assess advanced knowledge and reasoning abilities of foundation models *in a Chinese context*. C-EVAL comprises multiple-choice questions across four difficulty levels: middle school, high school, college, and professional. The questions span 52 diverse disciplines, ranging from humanities to science and engineering. C-EVAL is accompanied by C-EVAL HARD, a subset of very challenging subjects in C-EVAL that requires advanced reasoning abilities to solve. We conduct a comprehensive evaluation of the most advanced LLMs on C-EVAL, including both English- and Chinese-oriented models. Results indicate that only GPT-4 could achieve an average accuracy of over 60%, suggesting that there is still significant room for improvement for current LLMs. We anticipate C-EVAL will help analyze important strengths and shortcomings of foundation models, and foster their development and growth for Chinese users.[1]

## 1   Introduction

Evaluation benchmarks are at the core role for AI development. While traditional NLP benchmarks were mostly designed to measure specific and relatively simple abilities, large language models (LLMs), or foundation models, have demonstrated various new capabilities and shifted the evaluation focus to more general and intricate skills, such as broad world knowledge and complex reasoning. To align with the new era of LLMs, new benchmarks are proposed recently to probe a diverse set of LLM abilities. For example, MMLU (Hendrycks et al., 2021a), BIG-bench (Srivastava et al., 2022), and HELM (Liang et al., 2022) benchmarks attempt to aggregate a wide range of NLP tasks for holistic evaluation. Some other benchmarks specifically focus on advanced LLM abilities that emerge with scale, such as reasoning (Cobbe et al., 2021), hard math problem-solving (Hendrycks et al., 2021b), and coding (Chen et al., 2021). While traditional NLP benchmarks are becoming obsolete, these new ones are extensively used in recent research to drive development of the latest LLMs (Taylor et al., 2022; Chowdhery et al., 2022; Hoffmann et al., 2022; Touvron et al., 2023; OpenAI, 2023).

However, these modern benchmarks primarily target English language, resulting in limited understanding of LLMs' capabilities in other languages. In this work, we focus on evaluating the advanced abilities of foundation models in a Chinese context, one of the most widely spoken language in the

---

[*]Equal Contribution. Full list of individual contributions is detailed in Appendix A.

[†]Correspondence to Junxian He <junxianh@cse.ust.hk>. Work done while affiliated with SJTU.

[1]The C-EVAL data and evaluation code are available at https://github.com/hkust-nlp/ceval.

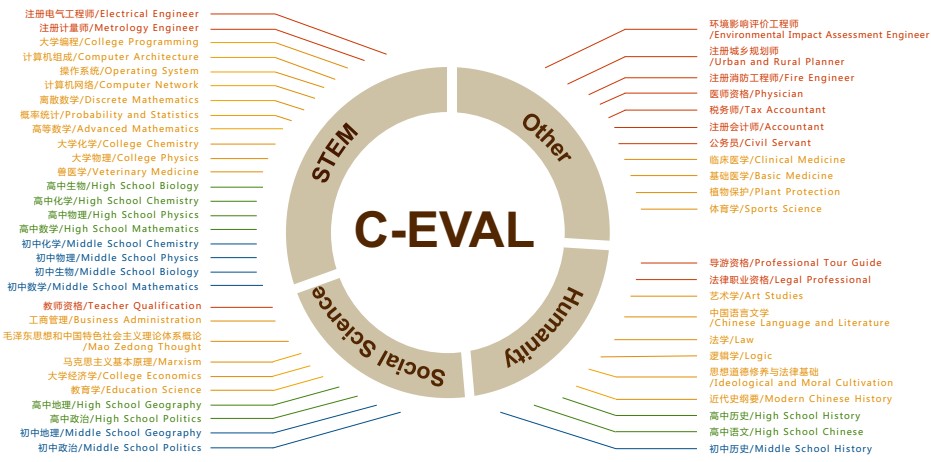

Figure 1: Overview diagram of C-Eval. Different colors of the subjects indicate four difficulty levels: middle school, high school, college, and professional.

world. Although there has been a recent surge in powerful Chinese LLMs, such as GLM-130B (Zeng et al., 2023), Wenxin Yiyan (Baidu, 2023), and MOSS (OpenLMLab, 2023), the corresponding evaluation significantly lags behind, with the CLUE benchmark (Xu et al., 2020), the Chinese counterpart of GLUE (Wang et al., 2019), serving as the best available standard. We emphasize that simply translating English benchmarks as in OpenAI (2023), even with flawless translations, does not fulfill the goal – LLMs intended for use in a Chinese environment should be evaluated on their knowledge of Chinese users' primary interests, such as Chinese culture, history, and laws, as well as other competencies unique in Chinese society. In contrast, English benchmarks tend to exhibit geographical biases towards the domestic knowledge of the regions that produce them.

To narrow the gap between Chinese LLM development and evaluation, we present C-Eval, the first comprehensive Chinese evaluation suite to thoroughly assess LLMs' advanced knowledge and reasoning abilities in a Chinese context. C-Eval consists of 13948 multiple-choice exam questions spanning 52 diverse disciplines, ranging from humanities to science and engineering, as depicted in Figure 1. The questions are collected from four difficulty levels: middle school, high school, college, and professional tests. Along with C-Eval, we introduce C-Eval Hard as an accompanied benchmark, a subset of particularly challenging subjects in C-Eval that demands highly advanced reasoning abilities to solve, such as advanced mathematics and college physics. Notably, C-Eval Hard is among the few benchmarks for *advanced* reasoning where GPT-4 still struggles, achieving an accuracy of 53.3%, making it the first Chinese benchmark at this level.

We conduct experiments to evaluate the most advanced LLMs on C-Eval in both answer-only and chain-of-thought settings. Results show that GPT-4 is the only model that surpasses 60% average accuracy. However, its 66.4% accuracy indicates that there is still large room for improvement in current LLMs. Despite not specially tailored for Chinese data, GPT-4, ChatGPT, and Claude emerge as the top three performers on C-Eval. Upon examining the results of LLMs focused on Chinese, we find that while some models managed to narrow the gap on Chinese knowledge test with ChatGPT, acquiring reasoning abilities seems more challenging. On C-Eval Hard, in particular, most models could only retain near-random accuracy. In addition to its use as a whole, we envision C-Eval as a suite of benchmarks, subsets of which could be separately utilized to assess certain model abilities of interest and analyze important strengths and limitations of foundation models. We hope C-Eval could guide the developers to understand the abilities of their models from multiple dimensions and facilitate the growth of foundation models for Chinese users.

## 2 The C-Eval Evaluation Suite

### 2.1 Design Principle

**Overview:** The motivation of C-Eval is to help developers quickly understand the abilities of their models from multiple dimensions, so that they could target the shortcomings of the models and

| Category | # Subjects | # Questions |
|---|---|---|
| *In terms of topic* | | |
| STEM | 20 | 4495 |
| Humanities | 11 | 2676 |
| Social Science | 10 | 2845 |
| Other | 11 | 3932 |
| *In terms of difficulty level* | | |
| Middle School | 7 | 1409 |
| High School | 8 | 1594 |
| College | 25 | 6249 |
| Professional | 12 | 4696 |
| *In terms of split* | | |
| Dev | 52 | 260 |
| Valid | 52 | 1346 |
| Test | 52 | 12342 |
| Total | 52 | 13948 |

Table 1: Statistics of C-EVAL.

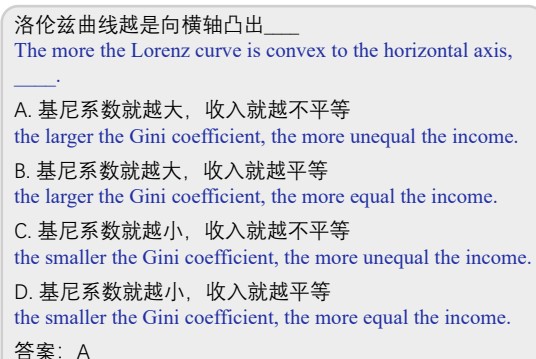

Figure 2: Example from college economics. English translations are shown for better readability.

improve them accordingly. To this end, we focus on the relatively advanced abilities of LLMs such as world knowledge and reasoning, which are arguably the most critical skills for LLMs nowadays. While different LLMs may perform similarly in simple scenarios like casual conversations, complex tasks are often the key differentiators between LLMs (OpenAI, 2023). Therefore, we construct C-EVAL from real-world, challenging human exams in China that are used to assess humans' abilities from multiple dimensions. We only select questions of a multi-choice format, similar to Hendrycks et al. (2021a), because: (1) metrics are clearly defined (i.e. accuracy), and (2) multi-choice questions are a simple but good proxy to evaluate the *potential* of advanced abilities of foundation models, which we consider could be easily exploited and reflected in various downstream applications through specialized instruction tuning (Chung et al., 2022; Wang et al., 2022). Each question has four choices and only one choice is the correct answer. LLMs are intended to be used to solve these questions through prompting. The questions in C-EVAL span 52 diverse disciplines that we later cluster them into broader categories as STEM, humanities, social science, and other areas. Summarized statistics of C-EVAL is shown in Table 1, and more detailed statistics per subject are in Appendix B.

**Attempt to mitigate data contamination:** Exam questions from national tests, such as China's national college entrance exams (commonly known as Gaokao) and national professional exams, are often widely distributed and accessible online. Consequently, these questions may inadvertently be crawled and incorporated into LLM pretraining, leading to potential data leakage issues. To mitigate this risk, we collect our data either from mock exams or from small-scale local exams, such as those available online from specific high schools. This deviates from previous work that built benchmarks using the exact questions from official national exams (Zhong et al., 2023). Moreover, most samples in C-EVAL are sourced from PDF or Microsoft Word documents on the Internet, rather than directly from plain text or structured questions. These documents are subsequently parsed and carefully annotated by the authors to obtain the final structured format, a process that often involves complex LaTeX equation conversion for certain subjects. This further minimizes the risk of data contamination.

## 2.2 Data Collection

**Subject selection:** C-EVAL covers four difficulty levels: middle school, high school, college, and professional. We include the standard subjects for middle and high school levels in China, except for the English subject.[2] For the college level, we select 25 representative subjects from all the 13 official categories of undergraduate majors listed by the Ministry of Education in China,[3] at least one subject from each category is included in C-EVAL to assure comprehensiveness. For the professional level, we refer to the official national vocational qualification directory in China[4] and choose 12

---

[2]We also exclude the middle school Chinese subject since the corresponding exam mostly focuses on writing responses to questions with few multi-choice questions.

[3]The Undergraduate Major Catalogue of Higher Institutions in China.

[4]National Vocational Qualifications Directory in China.

> 25 ℃时，将pH=2的强酸溶液与pH=13的强碱溶液混合，所得混合液的pH=11，则强酸溶液与强碱溶液的体积比是(忽略混合后溶液的体积变化)＿＿
>
> At 25 ℃, when a strong acid solution with pH=2 is mixed with a strong alkali solution with pH=13, the resulting mixture has a pH of 11. The volume ratio of the strong acid solution to the strong alkali solution is (ignoring any volume changes upon mixing) ＿＿.
>
> A. 11:1  B. 9:1  C. 1:11  D. 1:9
>
> 答案：B
> Answer: B
>
> 答案解析：
> 1. pH=13的强碱溶液中c($OH^-$)=0.1mol/L，pH=2的强酸溶液中c($H^+$)=0.01mol/L，酸碱混合后pH=11，即c($OH^-$)=0.001mol/L。
> 2. 设强酸和强碱溶液的体积分别为x和y，则:c($OH^-$)=(0.1y-0.01x)/(x+y)=0.001，解得x:y=9:1。
> Explanation:
> 1. In the strong alkali solution with pH=13, c($OH^-$) = 0.1mol/L, and in the strong acid solution with pH=2, c($H^+$) = 0.01mol/L. After mixing, the pH is 11, which means that c($OH^-$) = 0.001mol/L.
> 2. Assuming the volumes of the strong acid and strong alkali solutions are x and y, respectively, then: c($OH^-$) = (0.1y - 0.01x)/(x+y) = 0.001. Solving for x:y = 9:1.

Figure 3: An development example with explanations from high school chemistry. English translations are shown below the corresponding Chinese text for better readability.

representative ones, such as physician, legal professional, and civil servant qualification exams. We also cluster these subjects into four categories in terms of their topic: STEM (Science, Technology, Engineering, and Mathematics), Social Science, Humanities, and Other areas. All the 52 subjects and their assigned categories are illustrated in Figure 1.

**Data sources:** The primary source of our data is mock exams freely available on the internet. In addition, a portion of the college-level questions are past exam questions from top universities in China, publicly shared by the students. A minor fraction of college questions are mock questions for the national graduate entrance exam, sourced from the Weipu website[5] – these questions are not freely available to the public, and we have obtained their authorization to include around 2000 such questions into C-EVAL.

**Data Processing:** The collected data come in various formats, primarily as PDF or Microsoft Word documents and a minor fraction as web pages. PDF documents are initially processed into text. All questions are subsequently parsed – automatically when possible, and otherwise manually by the authors – into a structured format, as exemplified in Figure 2. For subjects with complex mathematical notations such as many subjects in the STEM category, we manually convert them into standard LaTeX formats, similar to Hendrycks et al. (2021b); Taylor et al. (2022). All the questions in C-EVAL are processed to include exactly four choices. Most of the original questions were accompanied by four choices already, and we eliminate questions with fewer than four options and randomly drop incorrect choices for questions with more than four options. All questions also go through the standard data preprocessing pipeline, such as deduplication and cleaning. Following this, the questions undergo several rounds of human validation by the authors, and all the LaTeX notations are ensured to be complied without syntax errors. We process at least around 200 questions for each subject, and randomly split the questions into a development set, a validation set, and a test set within each subject. The development split per subject consists of five exemplars to facilitate few-shot evaluation. These dev exemplars are also annotated with explanations to enable few-shot chain-of-thought settings (Wei et al., 2022), as we detail next. The validation and test set are created with a 1:9 ratio, where the validation set is intended to be used for hyperparameter tuning.

**Explanation data generation:** Chain-of-thought (COT) reasoning (Kojima et al., 2022; Wei et al., 2022) – that prompts LLMs to generate a text sequence of reasoning process along with the final answer – has shown great success on reasoning-heavy tasks. Compared to zero-shot COT, the few-shot version is more commonly used and achieves the state-of-the-art performance on various tasks (Gao et al., 2022; Wang et al., 2023; Zhou et al., 2023; Xie et al., 2023). To facilitate the potential usage of C-EVAL in a few-shot COT setting, we combine automatic generation and human annotation to produce high-quality explanation data for the development split. Specifically, we first prompt GPT-4 to generate step-by-step explanation to explain the ground-truth answer, then we

---

[5]https://kaoyan.cqvip.com/view/postgraduate/index.aspx

Figure 4: Example from advanced mathematics, a subject in C-EVAL HARD. English translations are shown below the corresponding Chinese text for better readability.

manually revise the generated explanations to obtain the final explanations. Details on prompting GPT-4 are in Appendix C. A dev example with explanations is illustrated in Figure 3.

### 2.3 C-EVAL HARD

We select 8 challenging math, physics, and chemistry subjects from C-EVAL to form a separate benchmark, C-EVAL HARD, which includes advanced mathematics, discrete mathematics, probability and statistics, college chemistry, college physics, high school mathematics, high school chemistry, and high school physics. These subjects often involve with complex LaTeX equations and require non-trivial reasoning abilities to solve. An example from advanced mathematics is shown in Figure 4. C-EVAL HARD aligns with recent efforts to create difficult benchmarks to assess advanced reasoning abilities (Hendrycks et al., 2021b; Suzgun et al., 2022), which are the key differentiators among various LLMs and could reflect LLMs' potential in general and complex scenarios. We emphasize that C-EVAL HARD is the first Chinese benchmark to provide highly complicated reasoning questions.

### 2.4 Evaluation

We use accuracy as the metric. While ground-truth labels of the development and validation splits are released, we keep the labels of the test split private. This is to ensure the fair use of the C-EVAL, as the C-EVAL data may unconsciously be included in pretraining data due to web crawling. Instead, users are required to submit their model predictions to https://cevalbenchmark.com to automatically obtain the test accuracy, where a public leaderboard is maintained. Users have the option to include their submission results in the live leaderboard, depending on their own preference.

## 3 Experiment

### 3.1 Setup

We evaluate LLMs in both zero- and five-shot settings on C-EVAL, where the five exemplars are from the development split. We adopt regular expressions to extract answer choices from the model responses, ensuring that we can successfully extract answers for nearly all cases. We report answer-only (AO) results on both zero- and five-shot settings and chain-of-thought (COT) results on the five-shot setting only, as we found that it was often difficult to extract the answer choices from zero-shot COT predictions where the generation does not follow specific patterns. Prompts of AO and COT are shown in Appendix D. We note that in the COT setting, the five-shot exemplars could exceed the maximum context length of some LLMs for certain subjects. In such cases, we dynamically reduce the number of exemplars to fit within the context window.

### 3.2 Models

To give a comprehensive view of the status of LLM in a Chinese language context, we evaluate 11 accessible top-performing LLMs that are able to process Chinese input, covering diverse organizations and varying in size, as shown in Table 2. ChatGPT (OpenAI, 2022) and GPT-4 (OpenAI, 2023) are the strongest GPT model variants from OpenAI. Claude (Anthropic, 2022), developed by Anthropic, is often considered comparable to ChatGPT. We evaluate both the Claude-v1.3 and Claude-instant-v1.0 variants, with Claude-instant being a lighter version. Bloomz-mt (Muennighoff et al., 2022) is based on the pretrained multilingual BLOOM model (Scao et al., 2022) with multitask prompted finetuning, thus is suitable for non-English languages. LLaMA (Touvron et al., 2023) is probably

| Model | Creator | #Parameters | Access |
|---|---|---|---|
| GPT-4 | OpenAI | *undisclosed* | API |
| ChatGPT | OpenAI | *undisclosed* | API |
| Claude-v1.3 | Anthropic | *undisclosed* | API |
| Claude-instant-v1.0 | Anthropic | *undisclosed* | API |
| Bloomz-mt | BigScience | 176B | Weights |
| LLaMA-65B | Meta | 65B | Weights |
| GLM-130B | Tsinghua | 130B | Weights |
| ChatGLM-6B | Tsinghua | 6B | Weights |
| Chinese-LLaMA-13B | Cui et al. | 13B | Weights |
| Chinese-Alpaca-13B | Cui et al. | 13B | Weights |
| MOSS | Fudan | 16B | Weights |

Table 2: Models evaluated in this paper.

| Model | STEM | Social Science | Humanities | Other | Average |
|---|---|---|---|---|---|
| Random | 25.0 | 25.0 | 25.0 | 25.0 | 25.0 |
| GPT-4 | 65.2 | 74.7 | 62.5 | 64.7 | 66.4 |
| ChatGPT | 49.0 | 58.0 | 48.8 | 50.4 | 51.0 |
| Claude-v1.3 | 48.5 | 58.6 | 47.3 | 50.1 | 50.5 |
| Bloomz-mt | 39.1 | 53.0 | 47.7 | 42.7 | 44.3 |
| GLM-130B | 36.7 | 55.8 | 47.7 | 43.0 | 44.0 |
| Claude-instant-v1.0 | 38.6 | 47.6 | 39.5 | 39.0 | 40.6 |
| ChatGLM-6B | 33.3 | 48.3 | 41.3 | 38.0 | 38.9 |
| LLaMA-65B | 32.6 | 41.2 | 34.1 | 33.0 | 34.7 |
| MOSS | 31.6 | 37.0 | 33.4 | 32.1 | 33.1 |
| Chinese-Alpaca-13B | 27.4 | 39.2 | 32.5 | 28.0 | 30.9 |
| Chinese-LLaMA-13B | 28.8 | 32.9 | 29.7 | 28.0 | 29.6 |

Table 3: Zero-shot average accuracy (%) in answer-only setting. We report the average accuracy over the subjects within each category. "Average" column indicates the average accuracy over all the subjects.

the best open-weight foundation model so far that achieves the highest accuracy on the English MMLU benchmark within open-weight models. The aforementioned models except Bloomz-mt are English-oriented during development, while they are able to process Chinese input because a minor fraction of Chinese text is present in the pretraining data.

We further include recent LLMs developed by Chinese institutions or individuals that is Chinese-oriented. GLM-130B (Zeng et al., 2023) and ChatGLM-6B (THUDM, 2023a) are based on the General Language Model architecture (GLM, Du et al. (2022)) trained on English and Chinese data. ChatGLM-6B is further adapted on conversational data. Chinese-LLaMA (Cui et al., 2023) is an adaptation of LLaMA, which is further pretrained on Chinese data. Chinese-Alpaca (Cui et al., 2023) performs instruction tuning based on Chinese-LLaMA. MOSS (OpenLMLab, 2023) is the first publicly available Chinese LLM, and it follows a training procedure similar to ChatGPT. We note that there are some other commercial Chinese-oriented LLMs whose weights and APIs are not directly open to the public at the time of writing this paper, such as Wenxin Yiyan (Baidu, 2023), Tongyi Qianwen (Alibaba, 2023), and Xunfei Xinghuo (iFLYTEK, 2023), these models may have strong performance, yet we are not authorized to evaluate and publicize their results. Therefore, we only report results from models with open APIs or weights in this work, while we anticipate the developers of other models to submit and optionally publicize their models' results in our website. A detailed description of the evaluated models can be found in Appendix E.

## 3.3 Results

**General comparison:** Zero- and five-shot answer-only results are shown in Table 3 and Table 4 respectively. We report the average accuracy, while a detailed breakdown of accuracy per subject is provided in Appendix F. GPT-4 is the only model that exceeds 60% average accuracy, highlighting the challenge presented by C-EVAL. GPT-4 significantly outperforms all other models, with the second-best model, ChatGPT, trailing over 14 percentage points behind in both zero- and five-shot settings. Claude-v1.3 achieves similar performance to ChatGPT, in terms of both the category-wise

| Model | STEM | Social Science | Humanities | Other | Average |
|---|---|---|---|---|---|
| Random | 25.0 | 25.0 | 25.0 | 25.0 | 25.0 |
| GPT-4 | 67.1 | 77.6 | 64.5 | 67.8 | 68.7 |
| ChatGPT | 52.9 | 61.8 | 50.9 | 53.6 | 54.4 |
| Claude-v1.3 | 51.9 | 61.7 | 52.1 | 53.7 | 54.2 |
| Claude-instant-v1.0 | 43.1 | 53.8 | 44.2 | 45.4 | 45.9 |
| GLM-130B | 34.8 | 48.7 | 43.3 | 39.8 | 40.3 |
| Bloomz-mt | 35.3 | 45.1 | 40.5 | 38.5 | 39.0 |
| LLaMA-65B | 37.8 | 45.6 | 36.1 | 37.1 | 38.8 |
| ChatGLM-6B | 30.4 | 39.6 | 37.4 | 34.5 | 34.5 |
| Chinese-LLaMA-13B | 31.6 | 37.2 | 33.6 | 32.8 | 33.3 |
| MOSS | 28.6 | 36.8 | 31.0 | 30.3 | 31.1 |
| Chinese-Alpaca-13B | 26.0 | 27.2 | 27.8 | 26.4 | 26.7 |

Table 4: Five-shot average accuracy (%) in answer-only setting. We report the average accuracy over the subjects within each category. "Average" column indicates the average accuracy over all the subjects.

| Model | STEM | Social Science | Humanities | Other | Average |
|---|---|---|---|---|---|
| Random | 25.0 | 25.0 | 25.0 | 25.0 | 25.0 |
| GPT-4 | 67.3 | 76.5 | 64.4 | 66.6 | 68.3 |
| Claude-v1.3 | 51.9 | 63.2 | 50.9 | 53.6 | 54.2 |
| ChatGPT | 47.8 | 58.3 | 47.7 | 48.5 | 50.0 |
| Claude-instant-v1.0 | 43.3 | 52.7 | 41.3 | 42.4 | 44.5 |
| ChatGLM-6B | 29.9 | 40.0 | 37.9 | 34.5 | 34.5 |
| MOSS | 27.3 | 38.1 | 33.6 | 29.4 | 31.2 |
| LLaMA-65B | 28.0 | 36.3 | 29.3 | 30.0 | 30.3 |
| GLM-130B | 24.8 | 33.1 | 30.8 | 30.0 | 28.8 |
| Chinese-LLaMA-13B | 20.5 | 30.5 | 28.2 | 27.1 | 25.4 |

Table 5: Five-shot average accuracy (%) in chain-of-thought setting. We report the average accuracy over the subjects within each category. "Average" column indicates the average accuracy over all the subjects. Bloomz-mt and Chinese-Alpaca-13B are excluded as they could not generate valid reasoning and thus fail to answer for many questions.

average and the overall average. In addition to average accuracy, Table 9 in Appendix F reveals that GPT-4 surpasses ChatGPT in almost every subject, indicating a comprehensive advantage. Among Chinese-oriented models, GLM-130B exhibits the best performance, ranking the fifth in terms of both zero- and few-shot performance, 7.0 and 14.1 points behind ChatGPT in zero- and five-shot settings respectively. We observe that smaller models, despite undergoing instruction tuning, still struggle to achieve a 40% accuracy. This contradicts recent assertions that a 10B-scale instruction-tuned model can achieve comparable performance to ChatGPT (Taori et al., 2023; Chiang et al., 2023) – we argue that while these models may perform well on simpler tasks, their inherent advanced abilities significantly lag behind when faced with more complex scenarios.

**Does few-shot prompting help?** Comparing Table 4 to Table 3, we find that while few-shot prompting helps many models achieve better results, it hurts performance of GLM-130B, Bloomz-mt, ChatGLM-6B, MOSS, and Chinese-Alpaca-13B. All of these models have undergone instruction tuning,[6] and we hypothesize that the accuracy drop is because that these models have not (appropriately) incorporated few-shot demonstrations into the instruction tuning stage, as emphasized in Chung et al. (2022), thus sacrificing few-shot in-context learning performance to obtain enhanced zero-shot instruction-following abilities.

**Does chain-of-thought prompting help?** The average accuracy in the COT setting is reported in Table 5, while Table 10 in Appendix F provides a detailed breakdown of the accuracy per subject. We exclude Bloomz-mt and Chinese-Alpaca-13B since these two models are unable to generate valid COT reasoning for a large portion of questions, failing to produce final answers. All models achieve comparable or lower average accuracy than in the answer-only setting. This suggests that

---

[6]GLM-130B incorporates instruction tuning in the pretraining stage.

| Model | Zero-shot AO | Five-shot AO | Five-shot COT |
|---|---|---|---|
| GPT-4 | 53.3 | 54.9 | 56.8 |
| Claude-v1.3 | 37.6 | 39.0 | 39.2 |
| ChatGPT | 36.7 | 41.4 | 35.0 |
| Claude-instant-v1.0 | 32.1 | 35.5 | 33.4 |
| Bloomz-mt | 30.8 | 30.4 | – |
| GLM-130B | 30.7 | 30.3 | 22.6 |
| LLaMA-65B | 29.8 | 31.7 | 21.4 |
| ChatGLM-6B | 29.2 | 23.1 | 26.1 |
| MOSS | 28.4 | 24.0 | 21.6 |
| Chinese-LLaMA-13B | 27.5 | 27.3 | 15.4 |
| Chinese-Alpaca-13B | 24.4 | 27.1 | – |

Table 6: Average accuracy on C-EVAL HARD in both answer-only (AO) and chain-of-thought (COT) settings.

COT prompting does not necessarily improve results for many subjects in C-EVAL. The primary reasons for this are twofold: (1) many subjects in C-EVAL are not reasoning-intensive, and additional reasoning steps may impair performance. This observation is supported by Chung et al. (2022), who noted performance degradation on MMLU with COT prompting. (2) Some models fail to leverage the benefits of COT prompting, particularly those that did not undergo COT-inclusive instruction tuning. Chung et al. (2022) reported this, noting an 8-point accuracy drop when using COT on MMLU with the 540B PaLM model. This finding partly elucidates the significant decrease in performance of the GLM-130B and LLaMA-65B models in the COT setting. Encouragingly, we still observe that COT prompting leads to considerable improvements for some models in certain subjects – for example, detailed results in Table 10 show that COT improves GPT-4's performance in college physics from 50.6% to 60.2%, in probability and statistics from 53.6% to 62.0%, ChatGLM's performance in middle school physics from 20.2% to 41.0%, and in high school geography from 29.2% to 38.2%.

**Difference between English- and Chinese-oriented models:**  GLM-130B is the best-performing Chinese-oriented model in our assessment, thus we focus on comparing it to the represented English-oriented model, ChatGPT, in zero-shot answer-only settings. We do not analyze GPT-4 here since it is not at the same level as all other models, and comparing GLM-130B to it is not very helpful and informative. As illustrated in Table 3, while GLM-130B underperforms ChatGPT by 7.0 points on overall average, the gap significantly narrows on the social science and humanities category, lagging only 2.2 and 1.1 points behind respectively. This reflects that by leveraging more Chinese data, the model might achieve performance on par with or even superior to ChatGPT in areas pertaining to Chinese knowledge, such as history, politics, and law, highlighting situations where Chinese-oriented models may have the upper hand. However, concurrently, we note a significant difference of 12.3 points between GLM-130B and ChatGPT in the STEM category, which implies a substantial gap in more complex tasks that necessitate advanced reasoning skills.

**Results on C-EVAL HARD:**  Table 6 shows the average accuracy on C-EVAL HARD. GPT-4 can only achieve 53.3%, 54.9%, 56.8% accuracy on zero-shot AO, five-shot AO, and five-shot COT settings respectively, implying the difficulty of C-EVAL HARD. Interestingly, chain-of-thought prompting improves GPT-4 slightly on these extremely challenging subjects. Indeed, only GPT-4, ChatGPT, and Claude manage to make meaningful progress – improving by at least 10 points – over a random baseline. Our results further confirm that some critical distinction among LLMs comes out when the tasks become complex enough. We underscore the importance of evaluating LLMs in such challenging settings, as current LLM development goes beyond creating a casual chatbot – it involves the development of complex systems or agents capable of interacting with various data types, receiving feedback, reasoning and using tools, and even performing actions (Mialon et al., 2023).

**Results on the validation split:**  Since we do not publicly release the labels for our test split, we provide the average accuracy on the validation split as a reference for developers. The validation split comprises a total of 1346 questions, with each subject contributing fewer than 30 validation questions on average. Therefore, tracking accuracy on a specific subject may not yield significant insights. Instead, we report the average answer-only accuracy across all subjects in Table 7. The average validation accuracy closely mirrors the average test accuracy as presented in Table 3 and Table 4. Additionally, the model ranking on the validation split broadly corresponds to that on the test split. These observations suggest that developers may use the average validation accuracy as a good indicator for expedited development processes.

| Model | Zero-shot | Five-shot |
|---|---|---|
| GPT-4 | 66.7 | 69.9 |
| Claude-v1.3 | 52.1 | 55.5 |
| ChatGPT | 50.8 | 53.5 |
| Bloomz-mt | 45.9 | 38.0 |
| GLM-130B | 44.2 | 40.8 |
| Claude-instant-v1.0 | 43.2 | 47.4 |
| ChatGLM-6B | 39.7 | 37.1 |
| LLaMA-65B | 38.6 | 39.8 |
| MOSS | 35.1 | 28.9 |
| Chinese-Alpaca-13B | 32.0 | 27.2 |
| Chinese-LLaMA-13B | 29.4 | 33.1 |

Table 7: Average accuracy on the validation split in the answer-only setting.

## 4 Related Work

**English benchmarks:** Traditional English benchmarks mainly focus on assessing certain abilities of models on a single task or a single type of tasks, such as natural language understanding (NLU, Wang et al. (2019)), reading comprehension (Rajpurkar et al., 2018), machine translation (Bojar et al., 2014), and summarization (Hermann et al., 2015; Narayan et al., 2018). As a representative example, the GLUE benchmark (Wang et al., 2019) combines a collection of NLU tasks, and has witnessed superhuman model performance due to the burst of pretraining models such as BERT (Kenton & Toutanova, 2019) and GPT (Radford et al., 2019). In order to assess the capabilities of LLMs more comprehensively, recent benchmarks have cast light on the broader knowledge and advanced abilities. The MMLU benchmark (Hendrycks et al., 2021a) provides multi-domain and multi-task evaluation collected from real-world examinations and books. LLMs' performance on MMLU fluctuates around random-chance accuracy until they reach the scale of GPT-3. The BIG-bench benchmark (Srivastava et al., 2022) consists of 204 diverse tasks, some of which are considered to be beyond the capabilities of current LLMs. The HELM benchmark (Liang et al., 2022) aggregates 42 different tasks and evaluates LLMs with 7 metrics ranging from accuracy to robustness.

**Chinese benchmarks:** Despite the flourishing of English benchmark, language abilities in Chinese language environment remain under-developed. The CLUE benchmark (Xu et al., 2020) is the first large-scale Chinese NLU benchmark, and still serves as the most widely-used and best available Chinese benchmark. Recently, the AGIEval benchmark (Zhong et al., 2023) contains data from the Chinese College Entrance Exam, Chinese lawyer qualification test and Chinese civil service examination. The MMCU benchmark (Zeng, 2023) consists of tests from four major domains including medicine, law, psychology and education, which are also collected from Chinese College Entrance Exam, qualification test as well as university examinations. Compared to AGIEval and MMCU, C-EVAL (1) has a broader coverage of domains (§2.2), (2) features four different levels of difficulty – particularly, the C-EVAL HARD benchmark is the first Chinese benchmark to provide sophisticated reasoning problems, and (3) makes an effort to mitigate data leakage – our questions mostly come from mock exams as PDF or Microsoft Word documents that are further processed by us, while AGIEval and MMCU collects the exact questions from past national exams in China.

## 5 Discussion

We believe that the evaluation of LLMs should transcend the scope of casual conversational bots, guiding developers in preparing LLMs to function in more complex scenarios. This was the primary motivation behind the creation of C-EVAL, a challenging evaluation suite. We hope that C-EVAL along with C-EVAL HARD have made important progress on this direction particularly in a Chinese context. We also note that, C-EVAL, along with all the English-language benchmarks, is far from perfect for LLM evaluation. There are many other abilities such as reasoning over and calling APIs, as well as multiple aspects that extend beyond accuracy, including safety, bias, and robustness. We leave further exploration on their evaluation for future work.

## Acknowledgement

We thank the anonymous reviewers for their comments. Yuzhuo Bai is supported by the National Key R&D Program of China (No. 2020AAA0106502) and Institute Guo Qiang at Tsinghua University.

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

## A    Author Contributions

**Data collection, annotation, and initial validation:**    Yuzhen Huang, Yuzhuo Bai, Zhihao Zhu, Junlei Zhang, Jinghan Zhang, Tangjun Su, Junteng Liu, Yikai Zhang, and Jiayi Lei collected the first version of the data, including some necessary manual annotation from PDF or Word documents. They then cross-validated the collected data. Junlei Zhang prompted GPT-4 to generate the explanation data, and then Yuzhen Huang, Yuzhuo Bai, and Junxian He manually revised the explanations.

**Data processing:**    Yuzhen Huang did most of the data processing job including a lot of manual revisions, such as fixing typos/incorrect formats of the questions from PDF documents and ensuring all the LATEX notations can be correctly compiled. Jinghan Zhang did the data deduplication.

**Evaluation:**    Yuzhen Huang tested ChatGPT and MOSS; Yuzhuo Bai tested ChatGLM; Junlei Zhang tested GLM-130B; Chuancheng Lv tested Bloomz-mt, LLaMA, Chinese-LLaMA, and Chinese-Alpaca; Yao Fu tested GPT-4 and the Claude model variants.

**Website and submission system:**    Zhihao Zhu built the website and the online submission system.

**Paper Writing:**    Yuzhen Huang, Yuzhuo Bai, and Junxian He wrote the main content of this paper, while other authors helped proofread.

**Advising:**    Yao Fu, Maosong Sun, and Junxian He take advisor roles in this project. Junxian He is the main advisor, initializing and organizing this project.

## B    Detailed Stats of C-EVAL

Table 8 lists all the C-EVAL tasks and their broader categories, as well as the number of questions included in each task.

## C    Explanation Data Generation

We show an example of the automatic COT explanation generation via GPT-4 in Figure 5. We use five human-written question-explanation pairs to prompt GPT-4 to generate explanation giving question and its correct answer. The generated explanations are further revised manually to obtain the final explanations.

## D    Evaluation Prompts

We show the evaluation prompts of answer-only and chain-of-thought test in Figure 6 and Figure 7 respectively.

## E    Details of the models being evaluated

**ChatGPT and GPT-4**    are GPT-series models that are enhanced to follow human instructions and be more helpful, harmless and honest using Reinforcement Learning from Human Feedback. GPT-4 additionally enables image inputs and goes through well-designed post-training alignment process, as well as having a larger scale than most of the existing model. GPT-4 achieves human-level performance on various benchmark, and even scored to be the top 10% in some simulated exams.

**Claude**    is the latest Anthropic-series LLM that also focuses on human intention alignment. Applying the constitutional AI approach(Bai et al., 2022), Claude manages to be both helpful and trustworthy. Claude-instant is the lighter vesrion with less cost and faster inference of Claude.

**BLOOMZ-mt**    (Muennighoff et al., 2022) is created by combining multitask prompted finetuning to the pretrained multilingual BLOOM model(Scao et al., 2022), using not only English prompts but also machine-translated prompts to match the language of multilingual tasks, and are found to be capable at task- and language-agnostic generalization. We evaluate the 176B version in our experiment.

| Subject | Category | # Questions |
|---|---|---|
| Advanced Mathematics (高等数学) | STEM | 197 |
| College Chemistry (大学化学) | STEM | 253 |
| College Physics (大学物理) | STEM | 200 |
| College Programming (大学编程) | STEM | 384 |
| Computer Architecture (计算机组成) | STEM | 219 |
| Computer Network (计算机网络) | STEM | 195 |
| Discrete Mathematics (离散数学) | STEM | 174 |
| Electrical Engineer (注册电气工程师) | STEM | 381 |
| High School Biology (高中生物) | STEM | 199 |
| High School Chemistry (高中化学) | STEM | 196 |
| High School Mathematics (高中数学) | STEM | 189 |
| High School Physics (高中物理) | STEM | 199 |
| Metrology Engineer (注册计量师) | STEM | 248 |
| Middle School Biology (初中生物) | STEM | 218 |
| Middle School Chemistry (初中化学) | STEM | 210 |
| Middle School Mathematics (初中数学) | STEM | 201 |
| Middle School Physics (初中物理) | STEM | 202 |
| Operating System (操作系统) | STEM | 203 |
| Probability and Statistics (概率统计) | STEM | 189 |
| Veterinary Medicine (兽医学) | STEM | 238 |
| Business Administration (工商管理) | Social Science | 339 |
| College Economics (大学经济学) | Social Science | 557 |
| Education Science (教育学) | Social Science | 304 |
| High School Geography (高中地理) | Social Science | 202 |
| High School Politics (高中政治) | Social Science | 200 |
| Mao Zedong Thought (毛泽东思想和中国特色社会主义理论体系概论) | Social Science | 248 |
| Marxism (马克思主义基本原理) | Social Science | 203 |
| Middle School Geography (初中地理) | Social Science | 125 |
| Middle School Politics (初中政治) | Social Science | 219 |
| Teacher Qualification (教师资格) | Social Science | 448 |
| Art Studies (艺术学) | Humanities | 336 |
| Chinese Language and Literature (中国语言文学) | Humanities | 237 |
| High School Chinese (高中语文) | Humanities | 202 |
| High School History (高中历史) | Humanities | 207 |
| Ideological and Moral Cultivation (思想道德修养与法律基础) | Humanities | 196 |
| Law (法学) | Humanities | 250 |
| Legal Professional (法律职业资格) | Humanities | 243 |
| Logic (逻辑学) | Humanities | 231 |
| Middle School History (初中历史) | Humanities | 234 |
| Modern Chinese History (近代史纲要) | Humanities | 240 |
| Professional Tour Guide (导游资格) | Humanities | 300 |
| Accountant (注册会计师) | Other | 497 |
| Basic Medicine (基础医学) | Other | 199 |
| Civil Servant (公务员) | Other | 481 |
| Clinical Medicine (临床医学) | Other | 227 |
| Environmental Impact Assessment Engineer (环境影响评价工程师) | Other | 317 |
| Fire Engineer (注册消防工程师) | Other | 318 |
| Physician (医师资格) | Other | 497 |
| Plant Protection (植物保护) | Other | 226 |
| Sports Science (体育学) | Other | 204 |
| Tax Accountant (税务师) | Other | 497 |
| Urban and Rural Planner (注册城乡规划师) | Other | 469 |

Table 8: Summary of all 52 subjects.

**LLaMA** (Touvron et al., 2023) is a Transformer-architecture LLM that is trained on a mixture of several open sources. Applying several improvements on vanilla Transformers used by previous LLMs, and optimized to improve the training efficiency, LLaMA shows strong language abilities and can surpass models having 10x larger parameters than LLaMA. We evaluate the LLaMA-65B version in our experiment.

**GLM-130B and ChatGLM-6B** are based on the General language model (GLM) structure that gain benefits from its bidirectional attention advantage. By using alterable number and length of

Figure 5: An example of generating explanations via GPT-4. The red text is the autocompleted response from model, while the preceding text is the user-inputted prompt. We indicate English translation below the corresponding Chinese text for each paragraph.

Figure 6: An example of few-shot evaluation in answer-only scenarios, while zero-shot evaluation is similar by removing the exemplars. The red text is the autocompleted response from model, while the preceding text is the inputted prompt. We indicate English translation below the corresponding Chinese text.

blanks, GLM can adapt to various tasks. GLM-130B is a bilingual pre-trained GLM that ultilizes self-supervised learning and multitask instruction pretraining. GLM-130B also realized INT4 quantization

Figure 7: An example of few-shot evaluation in chain-of-thought scenarios, while zero-shot evaluation is similar by removing the exemplars. The red text is the autocompleted response from model, while the preceding text is the inputted prompt. We indicate English translation below the corresponding Chinese text.

with little to no quality degradation that significantly accelerate its inference efficiency. ChatGLM-6B is a lightweight conversational version of the GLM family that has been specially optimized for Chinese contexts. ChatGLM-6B also applies quantization so it can be deployed with a consumer-grade graphic memory requirement as little as 6GB. We evaluate on the fp16 settings for both two models in our experiment.

**Chinese-LLaMA** is an adaptation of original LLaMA into Chinese language environments. Chinese-LLaMA expands the original LLaMA by adding 20K Chinese tokens into its vocabulary, and is secondarily pre-trained and instruction fine-tuned on Chinese data. We evaluate Chinese-LLaMA-13B in our experiment, the largest Chinese-LLaMA variant.

**Chinese-Alpaca** is based on the Chinese-LLaMA checkpoint that is further tuned on Chinese instruction tuning data. We evaluate Chinese-Alpaca-13B in our experiment, the largest Chinese-Alpaca variant.

**MOSS** is the first open-source Chinese LLM that matchs ChatGPT on both the training scale and alignment techniques. MOSS is initialized with CodeGen(Nijkamp et al., 2022), being pretrained on 100B Chinese tokens and 20B English tokens, and has further integrated supervised fine-tuning and preference model, as well as plugin augmentation, but not all the version are publicly available. We evaluate the moss-moon-003-sft version in our experiment.

# F    Breakdown of Model Performance

Table 9 and Table 10 show the detailed breakdown of accuracy per subject of four representative models in zero- and five-shot settings respectively, while we refer the readers to our website leaderboard https://cevalbenchmark.com/static/leaderboard.html for a detailed breakdown of results for all models.

| Subject | GPT-4 | ChatGPT | GLM-130B | Claude-instant-v1.0 |
|---|---|---|---|---|
| Advanced Mathematics | 48.6 | 38.2 | 28.9 | 36.4 |
| College Chemistry | 54.0 | 36.2 | 29.0 | 25.9 |
| College Physics | 50.6 | 34.1 | 29.5 | 33.5 |
| College Programming | 72.2 | 56.1 | 33.3 | 40.6 |
| Computer Architecture | 73.1 | 62.7 | 38.3 | 42.5 |
| Computer Network | 74.9 | 60.8 | 36.3 | 44.4 |
| Discrete Mathematics | 62.1 | 37.9 | 34.6 | 28.8 |
| Electrical Engineer | 48.7 | 38.9 | 31.6 | 32.7 |
| High School Biology | 69.1 | 49.1 | 37.7 | 40.0 |
| High School Chemistry | 55.2 | 45.3 | 36.0 | 39.5 |
| High School Mathematics | 41.0 | 28.3 | 34.9 | 24.7 |
| High School Physics | 65.1 | 36.6 | 26.9 | 40.0 |
| Metrology Engineer | 68.0 | 58.4 | 49.8 | 48.4 |
| Middle School Biology | 88.0 | 68.2 | 51.6 | 46.4 |
| Middle School Chemistry | 82.7 | 63.2 | 45.9 | 47.0 |
| Middle School Mathematics | 65.5 | 40.1 | 32.2 | 34.5 |
| Middle School Physics | 80.9 | 61.2 | 41.6 | 48.9 |
| Operating System | 79.3 | 70.4 | 42.5 | 46.4 |
| Probability and Statistics | 50.0 | 36.7 | 25.9 | 27.7 |
| Veterinary Medicine | 75.7 | 56.7 | 47.6 | 43.8 |
| Business Administration | 63.1 | 48.2 | 43.2 | 40.2 |
| College Economics | 67.6 | 47.3 | 38.6 | 40.8 |
| Education Science | 67.4 | 54.8 | 52.6 | 43.0 |
| High School Geography | 73.6 | 55.6 | 51.1 | 42.7 |
| High School Politics | 74.4 | 50.0 | 45.5 | 37.5 |
| Mao Zedong Thought | 74.9 | 56.6 | 67.6 | 49.3 |
| Marxism | 77.7 | 70.9 | 69.3 | 62.0 |
| Middle School Geography | 78.7 | 57.4 | 58.3 | 49.1 |
| Middle School Politics | 85.5 | 69.4 | 61.1 | 57.0 |
| Teacher Qualification | 84.5 | 69.9 | 70.7 | 54.9 |
| Art Studies | 57.4 | 47.7 | 49.3 | 33.2 |
| Chinese Language and Literature | 61.2 | 51.7 | 46.4 | 35.9 |
| High School Chinese | 38.8 | 37.6 | 25.8 | 31.5 |
| High School History | 66.5 | 48.4 | 48.4 | 41.2 |
| Ideological and Moral Cultivation | 73.8 | 65.1 | 59.3 | 58.1 |
| Law | 56.1 | 42.1 | 37.6 | 34.8 |
| Legal Professional | 55.3 | 40.5 | 35.8 | 36.7 |
| Logic | 62.3 | 38.7 | 34.3 | 36.3 |
| Middle School History | 83.6 | 65.2 | 65.2 | 48.8 |
| Modern Chinese History | 63.7 | 46.2 | 59.9 | 42.5 |
| Professional Tour Guide | 68.8 | 53.4 | 63.2 | 35.0 |
| Accountant | 63.4 | 46.3 | 39.7 | 38.1 |
| Basic Medicine | 78.9 | 61.1 | 46.9 | 39.4 |
| Civil Servant | 62.2 | 45.5 | 45.2 | 36.1 |
| Clinical Medicine | 70.5 | 51.0 | 43.5 | 38.0 |
| Environmental Impact Assessment Engineer | 59.4 | 52.0 | 45.6 | 41.3 |
| Fire Engineer | 45.0 | 41.5 | 35.8 | 29.4 |
| Physician | 75.6 | 59.4 | 44.7 | 41.5 |
| Plant Protection | 71.4 | 56.3 | 49.7 | 48.2 |
| Sports Science | 70.0 | 53.9 | 44.4 | 40.6 |
| Tax Accountant | 56.0 | 38.4 | 33.9 | 38.1 |
| Urban and Rural Planner | 59.1 | 49.0 | 43.5 | 38.0 |

Table 9: Zero-shot answer only accuracy per subject.

# G   Option Bias

The distribution of the correct answer is shown in the table 11. We admit that there are fluctuations in the proportion of different options. However, these fluctuations are relatively small around a random 25% level and similar to MMLU.

To verify if the option bias exists on models, we permuted the order of the choices in the questions and randomly selected 5 distinct permutations of choices for each question. Then we evaluate ChatGPT, ChatGLM-6B and ChatGLM2-6B (THUDM, 2023b) on the 5 different permutations of C-EVAL. The zero-shot results in the answer-only setting are shown in table 12. We report the average accuracy

| Subject | GPT-4 | ChatGPT | ChatGLM | Claude-instant-v1.0 |
|---|---|---|---|---|
| Advanced Mathematics | 49.7 / 53.8 | 48.0 / 29.5 | 3.5 / 23.1 | 39.9 / 41.6 |
| College Chemistry | 59.4 / 55.8 | 39.7 / 33.9 | 17.0 / 28.6 | 33.5 / 32.1 |
| College Physics | 50.6 / 60.2 | 35.8 / 35.2 | 14.8 / 30.1 | 35.8 / 31.3 |
| College Programming | 78.1 / 77.5 | 57.9 / 57.9 | 18.7 / 24.0 | 48.5 / 50.3 |
| Computer Architecture | 74.6 / 75.6 | 68.9 / 63.7 | 30.1 / 27.5 | 52.3 / 50.8 |
| Computer Network | 77.8 / 77.2 | 58.5 / 59.6 | 36.3 / 28.1 | 46.8 / 53.8 |
| Discrete Mathematics | 66.7 / 62.7 | 43.1 / 30.7 | 17.0 / 26.1 | 32.7 / 33.3 |
| Electrical Engineer | 49.9 / 56.0 | 39.5 / 44.0 | 23.9 / 31.3 | 31.3 / 35.7 |
| High School Biology | 72.6 / 70.3 | 53.7 / 48.0 | 28.0 / 26.9 | 44.6 / 38.9 |
| High School Chemistry | 52.3 / 55.8 | 52.9 / 36.0 | 19.8 / 28.5 | 39.0 / 37.8 |
| High School Mathematics | 38.0 / 42.8 | 34.3 / 37.3 | 4.2 / 24.1 | 24.1 / 32.5 |
| High School Physics | 69.1 / 61.1 | 43.4 / 34.9 | 10.9 / 21.7 | 44.0 / 26.9 |
| Metrology Engineer | 70.3 / 71.2 | 63.0 / 59.8 | 41.1 / 48.4 | 50.7 / 55.3 |
| Middle School Biology | 91.2 / 86.5 | 67.7 / 63.5 | 30.7 / 38.0 | 55.2 / 57.3 |
| Middle School Chemistry | 81.1 / 71.4 | 71.4 / 60.0 | 36.8 / 42.2 | 58.4 / 59.5 |
| Middle School Mathematics | 62.7 / 66.7 | 44.6 / 42.4 | 3.4 / 20.3 | 28.8 / 33.3 |
| Middle School Physics | 81.5 / 78.7 | 66.9 / 57.9 | 20.2 / 41.0 | 52.8 / 53.4 |
| Operating System | 82.1 / 80.4 | 71.5 / 60.3 | 40.8 / 27.4 | 57.0 / 57.5 |
| Probability and Statistics | 53.6 / 62.0 | 33.7 / 42.8 | 27.7 / 26.5 | 34.9 / 31.9 |
| Veterinary Medicine | 80.0 / 80.0 | 64.3 / 58.1 | 39.5 / 35.2 | 51.0 / 53.3 |
| Business Administration | 67.1 / 65.4 | 52.8 / 46.8 | 32.6 / 34.9 | 44.2 / 42.9 |
| College Economics | 71.4 / 74.4 | 52.5 / 51.3 | 22.1 / 30.0 | 47.7 / 50.3 |
| Education Science | 68.5 / 69.3 | 59.6 / 54.4 | 37.8 / 33.0 | 51.1 / 51.9 |
| High School Geography | 74.7 / 70.2 | 58.4 / 55.1 | 29.2 / 38.2 | 52.8 / 49.4 |
| High School Politics | 77.8 / 69.3 | 52.3 / 51.1 | 25.0 / 29.6 | 38.1 / 39.8 |
| Mao Zedong Thought | 79.5 / 79.9 | 60.7 / 63.5 | 46.6 / 50.7 | 47.0 / 50.2 |
| Marxism | 81.6 / 82.7 | 71.5 / 70.4 | 40.2 / 49.2 | 66.5 / 63.7 |
| Middle School Geography | 83.3 / 83.3 | 60.2 / 56.5 | 33.3 / 44.4 | 62.0 / 52.8 |
| Middle School Politics | 87.1 / 85.5 | 74.1 / 67.4 | 41.5 / 40.9 | 60.6 / 64.3 |
| Teacher Qualification | 85.0 / 85.0 | 75.7 / 66.9 | 48.9 / 49.1 | 68.4 / 62.2 |
| Art Studies | 65.1 / 66.1 | 49.7 / 52.3 | 34.6 / 36.9 | 35.6 / 38.9 |
| Chinese Language and Literature | 61.2 / 67.0 | 50.2 / 51.7 | 32.5 / 37.3 | 41.2 / 44.0 |
| High School Chinese | 37.6 / 39.3 | 36.0 / 27.5 | 26.4 / 19.1 | 30.9 / 27.0 |
| High School History | 68.1 / 68.1 | 54.4 / 45.1 | 35.2 / 40.1 | 52.7 / 40.7 |
| Ideological and Moral Cultivation | 77.3 / 77.3 | 66.9 / 68.0 | 48.3 / 52.3 | 62.8 / 63.4 |
| Law | 60.6 / 54.8 | 43.9 / 34.8 | 23.5 / 29.9 | 38.0 / 33.0 |
| Legal Professional | 54.4 / 48.4 | 44.7 / 32.6 | 27.0 / 26.5 | 39.5 / 30.2 |
| Logic | 60.3 / 63.2 | 37.7 / 35.8 | 32.8 / 31.4 | 38.7 / 35.8 |
| Middle School History | 84.5 / 86.5 | 62.8 / 63.8 | 48.3 / 52.7 | 58.5 / 52.2 |
| Modern Chinese History | 68.9 / 67.0 | 52.8 / 51.4 | 36.3 / 39.2 | 44.8 / 44.8 |
| Professional Tour Guide | 71.8 / 70.3 | 61.3 / 62.0 | 44.0 / 51.9 | 43.6 / 44.0 |
| Accountant | 64.6 / 61.9 | 51.7 / 42.0 | 23.9 / 32.7 | 47.2 / 41.3 |
| Basic Medicine | 80.6 / 78.3 | 61.1 / 60.6 | 31.4 / 33.7 | 49.7 / 45.1 |
| Civil Servant | 62.5 / 59.0 | 46.9 / 43.1 | 24.5 / 29.6 | 42.0 / 41.3 |
| Clinical Medicine | 76.0 / 73.5 | 55.0 / 53.0 | 34.5 / 32.0 | 36.5 / 37.5 |
| Environmental Impact Assessment Engineer | 63.7 / 59.8 | 50.9 / 49.1 | 37.7 / 35.6 | 47.3 / 47.3 |
| Fire Engineer | 50.0 / 49.6 | 42.6 / 37.9 | 28.4 / 32.3 | 36.5 / 36.2 |
| Physician | 76.8 / 76.1 | 63.7 / 54.6 | 34.8 / 37.7 | 50.6 / 46.5 |
| Plant Protection | 78.9 / 80.4 | 65.8 / 57.8 | 43.2 / 36.2 | 56.3 / 50.3 |
| Sports Science | 72.2 / 70.0 | 58.3 / 50.6 | 42.2 / 41.1 | 45.6 / 46.7 |
| Tax Accountant | 58.0 / 58.2 | 42.0 / 35.7 | 16.3 / 30.9 | 42.7 / 31.8 |
| Urban and Rural Planner | 63.2 / 65.6 | 52.2 / 49.3 | 36.8 / 37.3 | 45.5 / 42.3 |

Table 10: Five-shot accuracy per subject. We report both the answer only (AO) and chain-of-thought (COT) accuracy in an AO / COT format.

| Option | C-Eval | MMLU |
|---|---|---|
| A | 22.9% | 23.1% |
| B | 26.0% | 24.7% |
| C | 26.4% | 25.5% |
| D | 24.7% | 26.7% |

Table 11: The distribution of the correct answer.

across 5 permutations and the variance of the overall accuracy. The results imply that the variance across different permutations is relatively small.

| Model | STEM | Social Science | Humanities | Other | C-Eval Hard | Average | Var |
|-------|------|----------------|------------|-------|-------------|---------|-----|
| ChatGPT | 48.1 | 58.1 | 48.6 | 49.9 | 36.3 | 50.5 | 0.6 |
| ChatGLM-6B | 33.4 | 47.5 | 41.9 | 37.5 | 28.7 | 38.8 | 0.7 |
| ChatGLM2-6B | 38.1 | 58.3 | 49.6 | 45.2 | 29.3 | 45.9 | 0.2 |

Table 12: Zero-shot average accuracy (%) in answer-only setting. We report the average accuracy over 5 permutation within each category and overall accuracy. "Average" column indicates the average accuracy over 5 permutation. "Var" column indicates the variance of the overall accuracy.

# H    Compute and Resources Used for Evaluation

During our experiments to evaluate different LLMs on C-EVAL, we utilize a cluster with 8 A100-80GB GPUs to run the inference for models with released weights, such resources are required due to deploying three large models – Bloomz-mt (176B), LLaMA-65B, and GLM-130B. In most cases the inference on C-EVAL is finished within one day. For models with API access, we just run the inference with CPUs which finishes mostly within one day as well.

