# OpenReview forum: "C-Eval: A Multi-Level Multi-Discipline Chinese Evaluation Suite for Foundation Models"
_NeurIPS.cc/2023/Track/Datasets_and_Benchmarks — NeurIPS 2023 Datasets and Benchmarks Poster_

### Official Review · Reviewer_ZHas · 2023-07-17
**Review for paper 303**

**Rating:** 6
**Confidence:** 4
**Clarity:** Yes, the paper is very clear and easy…

**Strengths:**

- The proposed benchmark is comprehensive for evaluating LLMs’ knowledge and reasoning abilities in a Chinese context. This benchmark has been widely used by many recent LLMs (according to their website).
- The benchmark setting is reasonable. All questions and answers are manually checked, which guarantee the accuracy of the benchmark.
- The paper is very clear. The experiments adequately validate the LLMs’ ability of ICL and CoT.


**Additional Feedback:**

None.

**Correctness:**

The evaluation methods and experiment design are appropriate and performed correctly.

**Documentation:**

The documentation is good.

**Ethics:**

I do not see any ethical concerns.

**Limitations:**

The limitations and and potential negative societal impact of their work are well addressed.

**Opportunities For Improvement:**

- It seems that this benchmark is a Chinese version of “MMLU”. The novelty and technical depth are limited.
- It is still unknown that whether the performance on C-Eval benchmark can well reflect the real performance of the LLM.


**Relation To Prior Work:**

The related work is well discussed.

**Summary And Contributions:**

This paper proposes a new Chinese benchmark C-Eval for large language models (LLMs). The benchmark is designed to assess advanced knowledge and reasoning abilities of foundation models. C-Eval comprises multiple-choice questions in four topics (STEM, Humanities, Social Science, and Other), which are similar to the MMLU benchmark. The benchmark data are manually collected from various mock exams available on the Internet. The math-related problems are carefully converted into standard latex format. A benchmark website is built for quickly evaluating LLMs’ performance by uploading answers. Overall, the paper is clear and easy to follow. The benchmark is comprehensive and easy to use.

---

> ### Author Response · Authors · 2023-08-21
>
> Thanks for your time and comments! We acknowledge that C-Eval is like a Chinese version of MMLU and we believe that it is a valuable contribution for development of Chinese-oriented LLMs. Regarding the connection between the results on C-Eval and the real performance of LLMs in practice, we think that this is an important open research problem for many benchmarks including C-Eval and we leave it as future work to investigate.

---

### Official Review · Reviewer_WJEA · 2023-07-20
**An inclusive evaluation benchmark of multiple-choice style for LLMs with Chinese language support**

**Rating:** 8
**Confidence:** 4
**Correctness:** Yes.
**Clarity:** Yes.

**Strengths:**

The benchmark is useful for the current situation that there's a lack of inclusive dataset for the evaluation of increasing releases of LLMs. The scores on the benchmark can reflect how a LLM performs on the wide range of topics in Chinese. The data sources used for the construction of the dataset is of high quality as the questions are mostly exam questions in real use cases.  The authors conducted careful preprocessings and manual annotations to secure the quality of the data.

**Additional Feedback:**

NA

**Documentation:**

Yes.

**Ethics:**

No.

**Limitations:**

Yes.

**Opportunities For Improvement:**

The multiple-choice style of the benchmark is still problematic in the following aspects:
[1] Is there Option Bias on CEVAL which might affect the evaluation results?
[2] If a model generates correct reasoning procedures but fails to choose the target option, do you think this as a model issue or benchmark issue?
[3] Using accuracy as the metric is not optimal if serious Option Bias exists on a model. The reviewer suggests to consider a permutation-invariant metric  that the average of accuracy over all permutations of choices of a question. And report the inconsistency across permutations.


**Relation To Prior Work:**

Yes.

**Summary And Contributions:**

C-Eval is an inclusive benchmark to evaluate LLMs with Chinese language support. The benchmark consists of multiple-choice questions with different difficulty levels. It also contains a separate hard subset which requires advanced reasoning abilities. The benchmark has been proven to be challenging for current LLMs and is useful for the development of LLMs supporting Chinese.

---

> ### Author Response · Authors · 2023-08-21
>
> Thanks for your encouraging review! We address your comments below.
>
> > Is there Option Bias on CEVAL which might affect the evaluation results
> This is a good point. The distribution of the correct answer is shown in the table below. We admit that there are fluctuations in the proportion of different options. However, these fluctuations are relatively small around a random 25% level and similar to MMLU [1].
>
> | Option | C-Eval | MMLU  |
> |--------|--------|-------|
> | A      | 22.9%  | 23.1% |
> | B      | 26.0%  | 24.7% |
> | C      | 26.4%  | 25.5% |
> | D      | 24.7%  | 26.7% |
>
> > If a model generates correct reasoning procedures but fails to choose the target option, do you think this as a model issue or benchmark issue?
>
> This is a good question. Such cases are observed during our evaluation, which means that the final result of the model does not conform to the reasoning process. We think this is largely a model issue since the model fails the query. From the benchmark side, while it is better to provide more fine-grained metrics to reward correct reasoning steps to some extent in such cases, there are no well-established, automatic methods to evaluate the correctness of reasoning procedures so far.
> > Using accuracy as the metric is not optimal if serious Option Bias exists on a model. The reviewer suggests to consider a permutation-invariant metric that the average of accuracy over all permutations of choices of a question. And report the inconsistency across permutations.
>
> Thanks for the advice! We conducted the suggested experiment in the rebuttal, where we permuted the order of the choices in the questions and we randomly selected 5 distinct permutations of choices for each question. Then we evaluate ChatGPT, ChatGLM-6B and ChatGLM2-6B on the 5 different permutations of C-Eval. The zero-shot results in the answer-only setting are shown in the table below. We report the average accuracy across 5 permutations and the variance of the overall accuracy (var). The results imply that the variance across different permutations is relatively small.
>
> | Model       | STEM | Social Science  | Humanities  | Other  | C-Eval Hard | Average | Var |
> |-------------|------|-----------------|-------------|--------|-------------|---------|-----|
> | ChatGPT     | 48.1 | 58.1            | 48.6        | 49.9   | 36.3        | 50.5    | 0.6 |
> | ChatGLM-6B  | 33.4 | 47.5            | 41.9        | 37.5   | 28.7        | 38.8    | 0.7 |
> | ChatGLM2-6B | 38.1 | 58.3            | 49.6        | 45.2   | 29.3        | 45.9    | 0.2 |
>
>
> [1] Hendrycks et al. Measuring Massive Multitask Language Understanding. ICLR 2021.

---

### Official Review · Reviewer_KZk1 · 2023-07-23
**This review summarizes the submission as a comprehensive Chinese evaluation suite called C-EVAL, designed to assess advanced knowledge and reasoning abilities of foundational models in a Chinese context. It also introduces C-EVAL HARD, an accompanying benchmark that challenges current models in advanced reasoning. The strengths of the submission include the wide range of domains covered, the consideration of problem difficulty, and the comparison of different model settings.**

**Rating:** 7
**Confidence:** 4

**Strengths:**

The submission proposes a suite of benchmarks in a Chinese context, covering a wide range of domains. The benchmark fully considers the difficulty of problems by selecting questions from different difficulty levels, making its evaluation results more reliable. I think it has high quality, considering the lack of a Chinese context benchmark and its wide range of domains, which is quite helpful for later researchers. Moreover, all questions are carefully collected and preprocessed, trying their best to minimize possible ethical and social implications. The models evaluated are comprehensive, including both general-purpose models and specially trained models for the Chinese context. This work also compares different settings, such as zero-shot and few-shot settings, answer-only, and COT settings.


**Additional Feedback:**

NA

**Clarity:**

The paper has a clear structure and sufficient figures. The abstract briefly summarizes their contribution, the introduction analyzes the background and significance of the work and the implementation and experiments of the work are clearly and detailed written.


**Correctness:**

The submission is a benchmark. It seems that using questions of a multi-choice format as datasets has some merits and performs correctly, but is using accuracy as the metric a good evaluation method? There may be some cases that models have wrong reasoning steps but correct answers or models are restricted by some prerequisite knowledge, especially for professional questions. Moreover, although authors have tried to minimize the impact of data contamination, it may still have some negative effects on the result. I think it would be better to combine accuracy with other metrics.


**Documentation:**

There is sufficient detail to support reproducibility.


**Ethics:**

I don't think there are any ethical concerns with the submission that warrant further discussion or review.


**Limitations:**

They address some but not very much about the limitations. I think whether a model knows prerequisite knowledge about one question matters. I asked some easy professional questions but GPT returns the wrong answer. As a result, I think if we only use accuracy as the metric, it is vital to consider the impact of some prerequisite knowledge on a question. Besides, the submission lacks some comparison between this work and previous work. And this work contains a wide range of domains whose subsets can be used separately for a specific domain. However, there may be cases that for one domain, a benchmark that focuses on that domain works better than this work does.


**Opportunities For Improvement:**

 Although this work contains a wide range of domains, there are still amounts of domains that are not included. I think it may be a great improvement to include more domains, helping researchers in more fields. Moreover, this work uses accuracy as the metric but lacks some detailed explanation for its adequacy and necessity. Besides, I think it is vital to discuss the influence of a problem's prerequisites on the accuracy of the problem. And it can add some comparison between this work and previous work, and the comparison in one specific domain between this work and a benchmark that focuses on that domain.


**Relation To Prior Work:**

It clearly discussed that previous contributions are mostly focused on the English context and some specific domains. But this work proposes a benchmark in the Chinese context and contains a wide range of domains with different difficulties in each domain. However, it lacks some comparison between this work and previous work.


**Summary And Contributions:**

This paper presents C-EVAL, a comprehensive Chinese evaluation suite designed to assess advanced knowledge and reasoning abilities of foundational models in a Chinese context. Along with C-EVAL, this paper introduces C-EVAL HARD as an accompanied benchmark, which is among the few benchmarks struggling GPT-4 in advanced reasoning. Experiments show that even most advanced LLMs have moderate performances on C-EVAL, indicating large room for improvement in current LLMs. Moreover, C-EVAL contains diverse domains' questions, ranging from humanities to science and engineering, whose subsets can be used separately to access specific model abilities.

---

> ### Author Response · Authors · 2023-08-21
>
> Thanks for your time and encouraging review! We respond to your comments below.
>
> > This work uses accuracy as the metric but lacks some detailed explanation for its adequacy and necessity.
>
> We acknowledge the limitation of using accuracy as the metric. We use accuracy as an indicator mainly for simplicity and effectiveness considerations – it is a well-defined metric and commonly used when assessing LLMs. We agree with the reviewer that accuracy could be not ideal in some cases, for example, when the reasoning steps are wrong but the answer is correct – we think that it is debatable whether it should be counted as correct or wrong in such cases, and more fine-grained evaluation is not feasible so far due to the challenging evaluation of intermediate reasoning steps. This is why other benchmarks such as GSM8K [1] and MATH [2] also use accuracy as the metric ignoring the correctness of the intermediate reasoning steps.
>
> > I think it is vital to discuss the influence of a problem's prerequisites on the accuracy of the problem.
>
> This is a good point. A problem’s prerequisites may have an influence on the accuracy. We will add discussion on this aspect in the next revision, and see whether we could have some empirical insights on the influence of the problem’s prerequisites.
>
> > And it can add some comparison between this work and previous work, and the comparison in one specific domain between this work and a benchmark that focuses on that domain.
>
> Thanks for the suggestion! Domain-specific benchmarks are actually not directly comparable to this paper since C-Eval presents a comprehensive evaluation suite that covers diverse domains. While we agree that selecting a subset of domain-specific subjects from C-Eval and comparing to previous domain-specific benchmarks could be beneficial and provide more insights, domain-specific evaluation is not the focus of this paper and we leave such comparison as future work when we study LLMs’ performance on specific domains.
>
> [1] Cobbe et al. Training verifiers to solve math word problems. Arxiv 2021.
> [2] Hendrycks et al. Measuring Mathematical Problem Solving With the MATH Dataset. NeurIPS 2021.

---

### Official Review · Reviewer_jtRh · 2023-08-01
**C-EVAL: A Multi-Level Multi-Discipline Chinese Evaluation Suite for Foundation Models**

**Rating:** 6
**Confidence:** 5
**Correctness:** yes, it is designed and written in a …
**Clarity:** yes

**Strengths:**

1-Comprehensive Evaluation Suite: The paper introduces C-EVAL, a comprehensive evaluation suite specifically designed to assess the advanced knowledge and reasoning abilities of large language models in a Chinese context. The suite covers multiple-choice questions across various difficulty levels and a wide range of disciplines, making it a valuable resource for evaluating the capabilities of different models.
2-Real-World Challenging Exams: C-EVAL is constructed using real-world, challenging human exams in China, which are used to assess human abilities across multiple dimensions. This ensures that the evaluation is relevant to real-world scenarios and provides a meaningful assessment of model performance in complex tasks.
3-Focus on Advanced Abilities: The paper emphasizes the evaluation of advanced abilities in large language models, such as world knowledge and reasoning, which are crucial for their practical applications. By focusing on these complex tasks, the authors provide a more in-depth analysis of the strengths and weaknesses of various models.
4-Mitigation of Data Contamination: The authors take precautions to mitigate potential data contamination issues by carefully sourcing the data from mock exams and small-scale local exams, rather than using exact questions from official national exams. This ensures the fairness and integrity of the evaluation process.
Overall, the paper's strengths lie in its comprehensive evaluation suite, focus on advanced abilities, careful data handling, evaluation of Chinese-oriented models, and assessment of challenging scenarios, all of which contribute to a valuable and rigorous evaluation of large language models.

**Additional Feedback:**

The problem is well explained and sufficient data and experiments are provided to cover the claim.

**Documentation:**

yes

**Limitations:**

yes

**Opportunities For Improvement:**

1-Limited Scope of Evaluation Suite: While C-EVAL is a valuable evaluation suite, it is limited to multiple-choice questions and may not fully capture the diverse range of tasks that large language models are expected to perform in real-world applications. Other NLP tasks, such as text generation, translation, and dialogue generation, are not included in the evaluation, which might overlook important aspects of model performance.

2-Focus on Advanced Abilities: The paper's emphasis on evaluating advanced knowledge and reasoning abilities of models is commendable, but it may overlook the performance of models in more basic tasks. Understanding a model's performance in simple scenarios is also important as it serves as a baseline for more complex tasks.

3-Limited Data Sources: While efforts are made to mitigate data contamination issues, the reliance on mock exams and small-scale local exams may not fully represent the diversity and complexity of real-world data. The exclusion of national exams' questions might miss out on certain crucial aspects of language understanding and reasoning.

4-Chinese-Oriented Model Evaluation: While evaluating Chinese-oriented models is essential for the Chinese language context, the exclusion of other widely used English-oriented models may limit the broader perspective of the comparison. Including more English-oriented models would provide a comprehensive view of the state of large language models in both English and Chinese.

While this work presents a comprehensive evaluation suite and provides valuable insights into the capabilities of large language models, it has limitations in terms of scope, data sources, evaluation scenarios, and metrics, which may impact the generalizability of the findings and understanding of model performance in real-world applications.

**Relation To Prior Work:**

yes

**Summary And Contributions:**

The paper presents C-EVAL, the first comprehensive Chinese evaluation suite designed to assess the advanced knowledge and reasoning abilities of large language models (LLMs) in a Chinese context. C-EVAL comprises multiple-choice questions across four difficulty levels, spanning 52 diverse disciplines. The evaluation of several LLMs on C-EVAL indicates that there is still significant room for improvement in current LLMs. The study also investigates the impact of few-shot prompting and chain-of-thought reasoning on model performance. Furthermore, the authors introduce C-EVAL HARD, a subset of challenging subjects that require advanced reasoning abilities to solve. Overall, the results highlight the need for improved NLP benchmarks to match the rapid development of large language models.

---

> ### Author Response · Authors · 2023-08-21
>
> Thank you for the effort in reviewing our work! We basically agree with the reviewer on most of the points. However, we think that most of them are beyond the scope of this paper and we view them as potential future work for developing more comprehensive Chinese evaluation benchmarks for LLMs, thanks for the advice! We provide a detailed response to your comments below.
>
>
> > Limited Scope of Evaluation Suite: While C-EVAL is a valuable evaluation suite, it is limited to multiple-choice questions and may not fully capture the diverse range of tasks that large language models are expected to perform in real-world applications.
>
> We acknowledge the limited scope of evaluation present by C-Eval. However, considering the diverse range of LLMs’ abilities, we believe that multiple evaluation benchmarks rather than one single benchmark may be more desirable to holistically evaluate LLMs’ expected performance in real-world applications –  for example, on the English side, MMLU [1] is commonly used for knowledge evaluation, GSM8K [2],  MATH [3], and BBH [4] serve reasoning evaluation purposes, AlpacaEval [5] and MT-Bench [6] assess instruction-following abilities, while AgentBench [7] is to evaluate tool usage and planning. We see C-Eval as an important step on the Chinese side, and we anticipate the emergence of more Chinese benchmarks with different focuses in the near future.
>
> > Focus on Advanced Abilities: The paper's emphasis on evaluating advanced knowledge and reasoning abilities of models is commendable, but it may overlook the performance of models in more basic tasks.
>
> This is a good point. Prior to the advent of LLMs, there existed a series of Chinese benchmarks tailored to basic and simpler tasks. However, there has been a noticeable gap when it comes to advanced Chinese benchmarks, which is our motivation to construct C-Eval. While we agree that integrating previous basic datasets into C-Eval could provide a more holistic view, we chose to concentrate on advanced abilities to provide a more focused understanding of the data that we crafted. It also resonates with the current trajectory of LLM development, where advanced benchmarks take the dominant role. This is, in part, because LLMs frequently exhibit minimal difference when assessed on basic tasks.
>
> > Limited Data Sources: While efforts are made to mitigate data contamination issues, the reliance on mock exams and small-scale local exams may not fully represent the diversity and complexity of real-world data. The exclusion of national exams' questions might miss out on certain crucial aspects of language understanding and reasoning.
>
> The mock exams included in C-Eval are closely related to the real exam – they have similar scope and difficulty, and are frequently used in practice to train human students. While it is possible that national exams’ questions are more representative from some aspects, we had to balance considerations between potential data contamination and the authoritative quality of the data.
>
>
> > Chinese-Oriented Model Evaluation: While evaluating Chinese-oriented models is essential for the Chinese language context, the exclusion of other widely used English-oriented models may limit the broader perspective of the comparison. Including more English-oriented models would provide a comprehensive view of the state of large language models in both English and Chinese.
>
> Thanks for the advice! We chose to include the most powerful English-oriented models in our evaluation, such as the proprietary models GPT4, ChatGPT, Claude, and open-source models LLaMA-65B, GLM-130B, and Bloomz-mt-176B. As we continue maintaining the C-Eval benchmark, we will try to include more English-oriented models and we expect the community to participate to report results from different models on C-Eval as well.
>
> [1] Hendrycks et al. Measuring Massive Multitask Language Understanding. ICLR 2021.
> [2] Cobbe et al. Training verifiers to solve math word problems. Arxiv 2021.
> [3] Hendrycks et al. Measuring Mathematical Problem Solving With the MATH Dataset. NeurIPS 2021.
> [4] Suzgun et al. Challenging BIG-Bench Tasks and Whether Chain-of-Thought Can Solve Them. Arxiv 2022.
> [5] Li et al. AlpacaEval: An Automatic Evaluator of Instruction-following Models. Arxiv 2023.
> [6] Zheng et al. Judging LLM-as-a-judge with MT-Bench and Chatbot Arena. Arxiv 2023.
> [7] Liu et al. AgentBench: Evaluating LLMs as Agents. Arxiv 2023.

---

### Official Review · Reviewer_UDRo · 2023-08-03
**A comprehensive and useful benchmarks for Chinese Large Language Models**

**Rating:** 7
**Confidence:** 4
**Correctness:** n/a
**Clarity:** n/a

**Strengths:**

1. The datasets are comprehensive with different difficulties and disciplines. The datasets also supports various test setups.

**Additional Feedback:**

n/a

**Documentation:**

n/a

**Limitations:**

while this paper introduces a valuable dataset, this work itself is lack of new insights.

**Opportunities For Improvement:**

n/a

**Relation To Prior Work:**

n/a

**Summary And Contributions:**

This paper presents C-EVAL, a Chinese benchmark for the evaluation of large language models. It comprises multiple-choice questions across four difficulty levels: middle school, high school, college, and professional, and span 52 diverse disciplines, ranging from humanities to science and engineering. All the questions in C-EVAL is carefully collected. I believe it will be a very informative  and important benchmark for future research. The authors also report the results of existing popular large language models on the proposed benchmark.

---

> ### Author Response · Authors · 2023-08-21
>
> Thank you for your time and encouraging review!

---

### Decision · Program_Chairs · 2023-09-22

**Decision:**

Accept (Poster)

**Comment:**

This paper presents a benchmark for analysing Chinese LLMs.  Generally, reviewers agree that the benchmark is comprehensive it seems likely this benchmark will push forward work in Chinese language modelling.  This should be an interesting addition to the Benchmarks track.